# Emerging variants develop total escape from potent monoclonal antibodies induced by BA.4/5 infection

Chang Liu [1,2,13], Raksha Das [2,13], Aiste Dijokaite-Guraliuc [2,13], Daming Zhou [1,3,12,13], Alexander J. Mentzer [2,4], Piyada Supasa[2], Muneeswaran Selvaraj[2], Helen M. E. Duyvesteyn[3], Thomas G. Ritter [4], Nigel Temperton [5], Paul Klenerman[4,6,7,8], Susanna J. Dunachie [4,6,9,10], Neil G. Paterson [11], Mark A. Williams [11], David R. Hall [11], Elizabeth E. Fry [3] ✉, Juthathip Mongkolsapaya [1,2,10] ✉, Jingshan Ren [3] ✉, David I. Stuart [1,3,11] ✉ & Gavin R. Screaton [1,2] ✉

The rapid evolution of SARS-CoV-2 is driven in part by a need to evade the antibody response in the face of high levels of immunity. Here, we isolate spike (S) binding monoclonal antibodies (mAbs) from vaccinees who suffered vaccine break-through infections with Omicron sub lineages BA.4 or BA.5. Twenty eight potent antibodies are isolated and characterised functionally, and in some cases structurally. Since the emergence of BA.4/5, SARS-CoV-2 has continued to accrue mutations in the S protein, to understand this we characterize neutralization of a large panel of variants and demonstrate a steady attrition of neutralization by the panel of BA.4/5 mAbs culminating in total loss of function with recent XBB.1.5.70 variants containing the so-called 'FLip' mutations at positions 455 and 456. Interestingly, activity of some mAbs is regained on the recently reported variant BA.2.86.

Since emerging in late 2019, severe acute respiratory syndrome coronavirus-2 (SARS-CoV-2) is estimated to have led to 774 million infections and 7 million deaths (https://covid19.who.int/). Effective vaccines and natural infection have led to increasing levels of immunity, greatly reducing mortality but not preventing infection. SARS-CoV-2, therefore, still causes significant mortality in high-risk groups such as the immunosuppressed or elderly who show reduced or absent responses to vaccination.

All SARS-CoV-2 vaccines are designed to induce an antibody response to the spike protein (S), which is delivered in a variety of formats; RNA, viral vector, protein, or inactivated virus[1–3]. Analysis of panels of S-specific mAbs by several laboratories has led to considerable understanding of the antigenic landscape of S and sites of mAb binding[4–7]. To date, two domains of the S1 region of the spike, N-terminal (NTD)[8] and receptor binding (RBD)[7,9], have been described as binding sites for potent neutralizing mAbs. Most potent anti-RBD

[1]Chinese Academy of Medical Science (CAMS) Oxford Institute (COI), University of Oxford, Oxford, UK. [2]Nuffield Department of Medicine, Centre for Human Genetics, University of Oxford, Oxford, UK. [3]Division of Structural Biology, Nuffield Department of Medicine, University of Oxford, The Wellcome Centre for Human Genetics, Oxford, UK. [4]Oxford University Hospitals NHS Foundation Trust, Oxford, UK. [5]Viral Pseudotype Unit, Medway School of Pharmacy, University of Kent and Greenwich Chatham Maritime, Kent ME4 4TB, UK. [6]Peter Medawar Building for Pathogen Research, Oxford, UK. [7]Nuffield Department of Clinical Neurosciences, University of Oxford, Oxford, UK. [8]NIHR Oxford Biomedical Research Centre, Oxford, UK. [9]Nuffield Department of Medicine, Centre for Tropical Medicine and Global Health, University of Oxford, Oxford, UK. [10]Mahidol-Oxford Tropical Medicine Research Unit, Bangkok, Thailand. [11]Diamond Light Source Ltd, Harwell Science & Innovation Campus, Didcot, UK. [12]Present address: College of Life Sciences, Zhejiang University, Hangzhou 310058, China. [13]These authors contributed equally: Chang Liu, Raksha Das, Aiste Dijokaite-Guraliuc, Daming Zhou. ✉e-mail: liz@strubi.ox.ac.uk; juthathip.mongkolsapaya@well.ox.ac.uk; ren@strubi.ox.ac.uk; dave@strubi.ox.ac.uk; gavin.screaton@medsci.ox.ac.uk

mAbs bind on or near the receptor binding motif[5], a small 25 amino acid patch at the tip of the RBD. mAbs binding here neutralize by blocking the binding of S to the SARS-CoV-2 cellular receptor angiotensin-converting enzyme 2 (ACE-2)[10]. A second group of anti-RBD antibodies, exemplified by S309, bind close to the N-linked glycan at residue N343, do not block interaction with ACE2 and may act to destabilize the S-trimer[11]. Potent anti-NTD mAbs bind to a so-called supersite on the NTD, do not block ACE2 interaction and their mechanism of action is not well understood[8]. All commercially developed mAbs to date target the RBD[12–14], whilst anti-NTD mAbs tend to be variant-specific due to extensive mutation of the supersite between variants.

Both the RBD and NTD are hot spots of mutation in SARS-CoV-2. RBD mutations can impart selective advantages to the virus, firstly, some, for example, the N501Y mutation found in the Alpha variant, increase the affinity to ACE2 and are believed to drive increased transmissibility[15]. Secondly, mutations in the RBD and NTD may lead to escape from neutralizing antibody responses[16]. Pervasive high levels of neutralizing antibodies in the general population generate intense selective pressure on the virus to break through pre-existing immunity[17]. Evolution of S has therefore been rapid, with many mutations mapping closely to the sites of interaction of potent mAbs in the RBD and NTD[16]. SARS-CoV-2, in a period of 4 years since its emergence, has evolved variants that escape all mAbs developed for clinical use.

In this study, we report the generation of a panel of mAbs from vaccinated volunteers who suffered breakthrough BA.4/5 infections. We characterize these structurally and functionally and document a steady attrition of activity to a succession of variants culminating in the loss of activity of the whole panel to some currently circulating strains.

## Results

### Generation of mAbs from BA.4/5 infection samples
Blood was taken from 11 triple vaccinated volunteers >23 days (median 38) after the PCR test confirmed SARS-CoV-2 BA.4 infection ($n = 3$) or BA.5 infection ($n = 8$). Focus reduction neutralization tests (FRNT) were performed on Victoria (an early pandemic strain), together with BA.2, BA.4 and BA.5 live virus, to select the highest titre samples for antibody production (Fig. 1a). Seven samples with the highest titres against BA.4 or BA.5 were used for mAb production.

PBMCs were stained with BA.4/5 S trimer (the S sequence is the same for BA.4 and BA.5) and single IgG positive memory B cells were sorted (Fig. 1b). Meanwhile, 4 samples were stained with an S trimer which we term BA.4+all, constructed to harbour additional mutations seen in recent sub-lineages (G339H, R346T, L368I, K444R, V445P, G446S, N450D, L452M, N460K, V483A, E484R, F486S, F490V and S494P) in a BA.4/5 background. From the selected cells, a degenerate PCR reaction was used to amplify heavy and light chains, which were assembled into an expression vector using Gibson assembly and the products expressed by transient transfection. Supernatants were tested for binding to full-length BA.4/5 or BA.4+all S trimer, BA.4/5 or BA.4+all RBD and BA.4/5 NTD. From 861 sorted cells, 442 antibodies were recovered, leading to the selection of 28 potent RBD or NTD binding mAbs (BA.4/5-34, 35 and 36 mAbs were derived from the BA.4+all sort) showing 50% focus reduction (by FRNT) of BA.5 virus <100 ng/ml (Figure S1). Heavy chain (HC) gene usage corresponded to: IGHV1-69 (5/28), IGHV3-9 (4/28), and public gene family IGHV3-53 (4/28) and IGHV3-66 (5/28) (Fig. 1c, Table S1). The level of somatic mutation was comparable to a previous set of antibodies we developed following BA.1 infection, significantly greater than seen in early pandemic mAbs (Fig. 1d). Six mAbs showed little or no ACE2 blocking ability (BA.4/5-3, 4, 12, 15, 20 and 33) (Fig. 1e).

### Cross-reactivity of anti-BA.4/5 antibodies
Pseudo-typed virus neutralization assays[18] were used to test the antibodies against 26 variants seen throughout the pandemic with particular emphasis on Omicron sub-lineages (Fig. 2). All potent BA.4/5 mAbs, except BA.4/5-33 and -36, cross-neutralize early pandemic pseudovirus Victoria (IC50 < 100 ng/ml) and may have been selected from B cell clones originally generated following vaccination (Fig. S2). BA.4/5-33 and −36 were the only two anti-NTD antibodies we isolated and were specific to BA.2 derived variants.

Most BA.4/5 mAbs showed >5-fold reduction of neutralization titre on at least one Omicron sub-lineage, compared to BA.4/5 (Fig. 2a). Against circulating variants isolated after the emergence of BA.4/5 the activities of these mAbs were completely knocked out for 12/28, 14/28, 18/28, and 16/28 on the variants BA.2.75.2, BQ.1.1, BN.1, and CH.1.1, respectively. Against the latest dominating variants (XBB and its sub-lineages), the activities of the mAbs were also severely impaired, with 18/28 knocked out by XBB, XBB.1 and XBB.1.5. The most striking knockout was observed on XBB.1.5.10 and XBB.1.5.70, which have F456L and L455F + F456L ('FLip') mutations, respectively, on the XBB.1.5 background (Fig. 2b). XBB.1.5.70 containing the FLip mutation led to knock out of activity of the few remaining antibodies retaining activity against XBB.1.5, including the broadly neutralizing mAbs BA.4/5-1 and BA.4/5-2 together with mAbs 22, 28, 31, 34 and 35, leading to knock out of activity of all 28/28 mAbs from the BA.4/5 panel.

BA.2.86 is a newly emerging variant first recognized in August 2023, compared to its closest ancestor, BA.2, it has 38 amino acid changes in S with 14 changes in RBD, including a deletion (ΔV483). The large number of changes in S has led to the concern that it may show a greater level of immune escape. Here we tested BA.2.86 neutralization by the BA.4/5 panel of mAbs, where activity was similar to XBB.1.5, likely because BA.2.86 lacks mutations at residues 455 and 456. However, mutation L455S has been observed in 856 reported BA.2.86 sequences and has been named JN.1. Neutralization assays show that JN.1 knocks out mAbs BA.4/5−8, −10, −28, and −35, which show neutralizing activity against BA.2.86.

### Structures of anti-BA.4/5 mAbs
To elucidate the binding mode and detailed interactions of the most broadly neutralizing anti-BA.4/5 mAbs, we determined crystal structures of complexes of Delta-RBD[19] with Fabs BA.4/5-1 and EY6A[20], Delta-RBD with Fabs BA.4/5-2 and Beta-49[21], Delta-RBD with Fab BA.4/5-9 and Delta-RBD with Fab BA.4/5-35 (Figs. 3, 4, and Table S2). Representative electron density is shown in Fig. S3.

*BA.4/5-1* (IGHV4-39) is an RBD binding antibody that, according to the visual analogy of the RBD to a human torso[7], attaches at the back of the RBD, with the HC binding at the top of the neck and the LC at the back of the left shoulder, making a large footprint of 1310 Å$^2$ (HC 670 Å$^2$ and LC 640 Å$^2$) (Figs. 3a, e, f, and 4a, g). This antibody heavily overlaps the ACE2 binding site; of the 35 RBD residues in the BA.4/5-1 footprint, 20 overlap with the ACE2 footprint. In line with this, a large number of mutation sites in the Omicron variants have either direct contact with or are on the footprint of BA.4/5-1, including 405, 408, 417, 476, 477, 484, 486, 490, 493-494, 498, 501 and 505, but interestingly most of these mutations have little or no effect on the neutralization potency of BA.4/5-1, which maintains quite broad cross-reactivity (Fig. 2a). RBD residues L455, F456, Y489 and Q493 make extensive hydrophobic interactions (≤4 Å) with CDR-H3, while F486, G476 and S477 contact CDR-L3 of BA.4/5-1. F486 also makes ring-stacking contacts with Y35 and Y60 from the framework regions of the HC (Fig. 3e, f). Mutation of residue F486 has been observed in recently identified SARS CoV-2 variants: F486V in BA.4/5 and BQ.1, F486S in BA.2.75.2 and XBB and F486P in BA.2.10.4 and XBB.1.5 (Fig. 2b). It appears that despite close interaction with residue 486, BA.4/5-1 can tolerate mutations of F486 showing only modest reduction in titres to some of the newly described variants compared to BA.4/5 (Fig. 2a). Neutralization titres to BS.1 (a rarely described variant, which has failed to achieve a major breakthrough) were reduced 32-fold compared to BA.4/5 or BA.2.12.1. BS.1 contains the unique mutation G476S (Fig. 2b) compared to the other

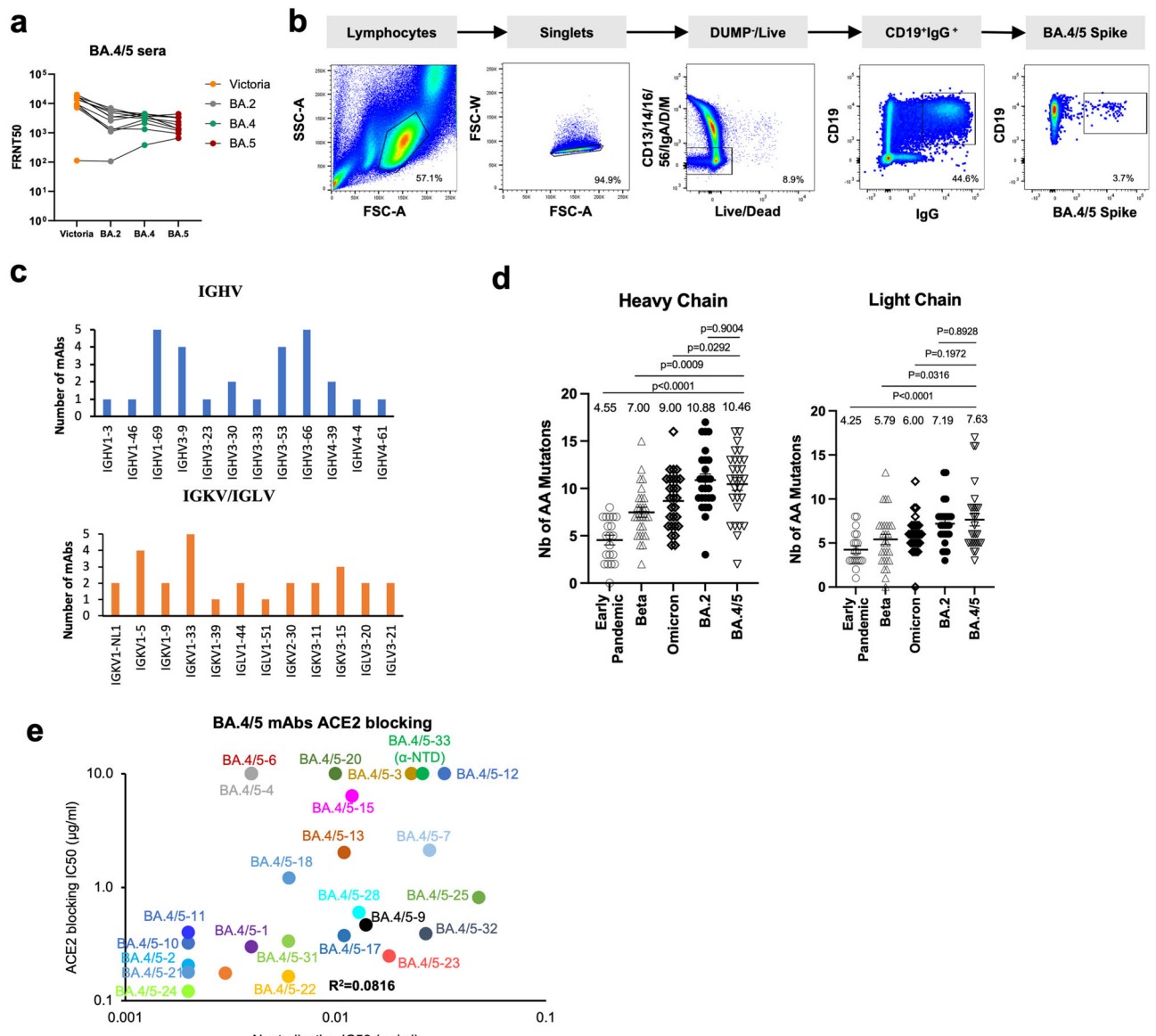

**Fig. 1 | Generation of BA.4/5 mAbs. a** Live virus neutralization FRNT50 titres against Victoria (an ancestral SARS-CoV-2 isolate), BA.2, BA.4 and BA.5 viruses using serum from vaccine breakthrough BA.4/5 serum. **b** Sorting strategy for BA.4/5 specific B cells. **c** Heavy chain and light chain gene usage of potent BA.4/5 mAbs. **d** Number of somatic mutations found in BA.4/5 mAbs ($n = 28$ samples) compared to sets previously isolated from early pandemic ($n = 20$ samples), Beta ($n = 27$ samples), BA.1 ($n = 28$ samples) and BA.2 ($n = 26$ samples) infection, which have been previously reported[36]. Mean with SEM are shown, and *p* values were calculated by Mann–Whitney test. **e** Blocking of ACE2-S interaction by potent BA.4/5 mAbs.

variants tested in Fig. 2a, G476 has close contact with Y92 of CDR-L3 and the change to Ser likely disrupts this interaction (Fig. 3f).

*BA.4/5-2* (IGVH3-30), a broadly neutralizing mAb, binds the RBD with the HC at the back of the left shoulder and a light chain at the back of the neck. The HC makes the most contact with the RBD, with a footprint of 650 Å$^2$ compared to 305 Å$^2$ for the LC (Figs. 3b, 4b, h). Of the 27 residues within this footprint, 12 overlap with the footprint of ACE2. CDR-H3 makes extensive interactions with RBD residues 416-417, 420-421, 455-460, 473 and 489. In contrast, CDR-L3 makes only weak contact with T415 of the RBD. CDR-H1 contacts residues 475-477 and 486-487 (Fig. 3g, h). Although residues 405, 408, 417, 460, 476-477 and 505, which are mutated in some recent variants (Fig. 2b), have direct contacts with BA.4/5-2, BA.4/5-2 retains the ability to broadly neutralize all the variants containing mutations at these residues except those also containing F456L or L455F and F456L mutations (Fig. 2a).

*BA.4/5-9* (IGVH1-46) is bound similarly to BA.4/5-1 with its HC sitting on top of the neck of the RBD but rotated by about 35° such that the LC is located at the back rather than at the back of the left shoulder as in BA.4/5-1, making a footprint of 1210 Å$^2$ (770 Å$^2$ by HC, 440 Å$^2$ by LC) (Figs. 3c, i, j, 4c, i). CDR-H3 contacts L455 and F456 through G101 and N103, while contacts to F456 from CDR-L3 are extensive, mainly from the main chain atoms of residues 92-94 and the Cβ of W94 (Fig. 3j). Interestingly, BA.4/5-9 is not sensitive to the F456L mutation of XBB.1.5.10 but is sensitive to L455S in JN.1 whilst the L455F and F456L double mutation knocks out the activity of the mAb. CDR-H2 is located directly on top of Q493 making a single contact through P53 to the latter, and the Q493R mutation found in BA.1, BA.2 and some sub-variants thereof diminishes the neutralization ability of BA.4/5-9. CDR-L1 interacts extensively with residues 415−416 and 420−421 none of which are in the ACE2 footprint and have not yet shown significant mutations.

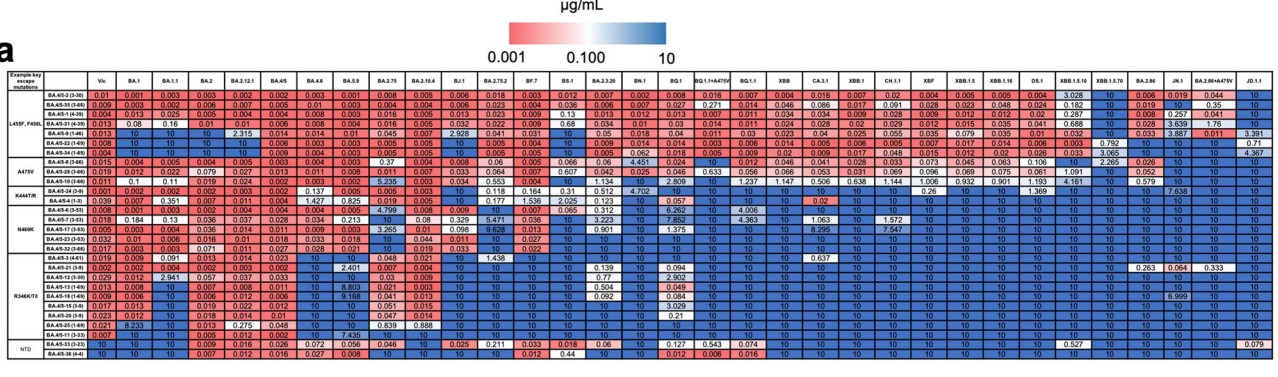

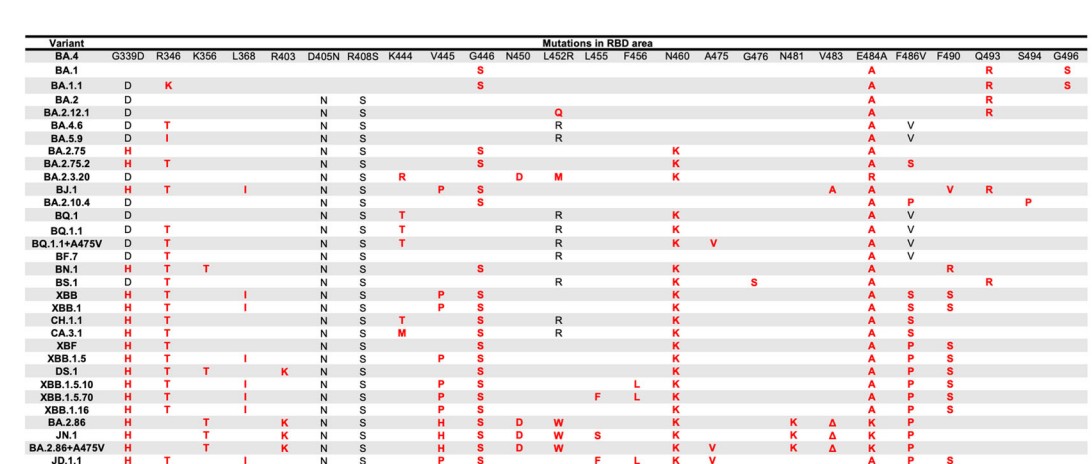

**Fig. 2 | Heatmaps of antibody IC50 neutralization titres and live virus-neutralizing activity of mAbs BA.4/5-2 and BA.4/5-5. a** Heatmap of IC50s of potent BA.4/5 mAbs against a panel of pseudoviruses expressing variant S sequences. All assays have been done at least twice. **b** Mutations in RBD for variants are shown in (**a**). Indicated mutations are additional to the S sequence of BA.4/5.

*BA.4/5-35* is highly unusual in that it belongs to the so-called public family of antibodies IGHV3-66, closely related to IGHV3-53, and binds a well-characterized epitope (Figs. 3d, k–m, 4d, j). Such antibodies were very common in early responses[5] and many escape mutations map to the shared epitope, leading to the general abolition of neutralization. The structure reveals how BA.4/5-35 dodges these mutations to remain highly cross-reactive. Of published Fabs in the PDB the BA.4/5-35 HC is most similar to Omi-3[22] (83% sequence identity for Vh). A substitution (D100G) in the HC creates space for the side chain of residue W94 of the light chain to stack against the peptide of HC residue 100. This leads to a significant refolding of the LC CDR3 and positions the whole LC further away from the RBD. The overall effect is that HC contacts are maintained, but the LC is lifted away from the escape mutations, conferring broad cross-reactivity (Fig. 3k–m). A recently described but low-frequency mutation in BQ.1.1, A475V, which maps to the BA.4/5-35 binding site, leads to a significant fall in activity.

### The L455F and F456L double "FLip" mutations knock out all BA.4/5 mAbs

Our structures show that the four most broadly neutralizing anti-BA.4/5 mAbs have substantial (12–21 residues) overlap with the ACE2 footprint. Mutations of many of these footprint residues (417, 445–446, 460, 477, 484, 486, 498, 501 and 505) have no effect on these mAbs, whilst other mutations show moderate effects (3- to 7-fold for G476S in BS.1 variant on BA.4/5-1, BA.4/5-2 and BA.4/5-35) or target just one of the mAbs, (A475V impacts BA.4/5-35 and Q493R knocks out BA.4/5-9). However, all four Fabs have close contact with residues 455 and 456 (Figs. 3, 4) and the L455F and F456L double "FLip" mutations, present in some of the recent fastest spreading variants, knock out all of them.

Whilst the F/L switch only requires a minimal third base switch in the codon (Phenylalanine: UUU or UUC, Leucine: UUA or UUG), the effect on the protein is a substantial change in side-chain volume, sufficient to disrupt surface complementarity between tightly interacting surfaces whilst maintaining hydrophobic properties. Hydrophobic surface patches are typically hallmarks of protein-protein interfaces and this region is central to the ACE2 footprint. Indeed, these mutations have also been reported to increase ACE2 affinity[23]. We note that IGHV3-53/66 mAbs[5] have become common among the potent antibodies induced from post-Omicron variant infections, e.g. 8 out of 28 BA.4/5 mAbs and 5 out of 10 XBB.1.5 mAbs. These all have close contact with 455 and 456. It appears that the L455F and F456L mutations have been generated under the pressure of a group of antibodies, including numerous IGHV3-53/66 mAbs, bound at the back of the RBD and that these mutations act in synergy[23].

### Discussion

Within 4 years, SARS-CoV-2 has concentrated remarkable mutational change in the S gene. In late 2021 Omicron marked a step change, with profound drops in neutralization titres in serum from vaccinees and from natural infection, leading to a global wave of infection and becoming the dominant variant in a matter of weeks. Since then, it has dominated and continued to evolve rapidly, BA.1 was replaced by BA.1.1 and BA.2, which were in turn replaced by BA.4/5, BA.4.6 and BF.7. Since late 2022, an increasing number of BA.2 sub-variants have begun to cocirculate, with convergent evolution leading to the acquisition of subsets of common mutations in related variants. In 2023, sub lineages related to XBB have been the dominant variants, although BA.2.86 has recently emerged, which has numerous

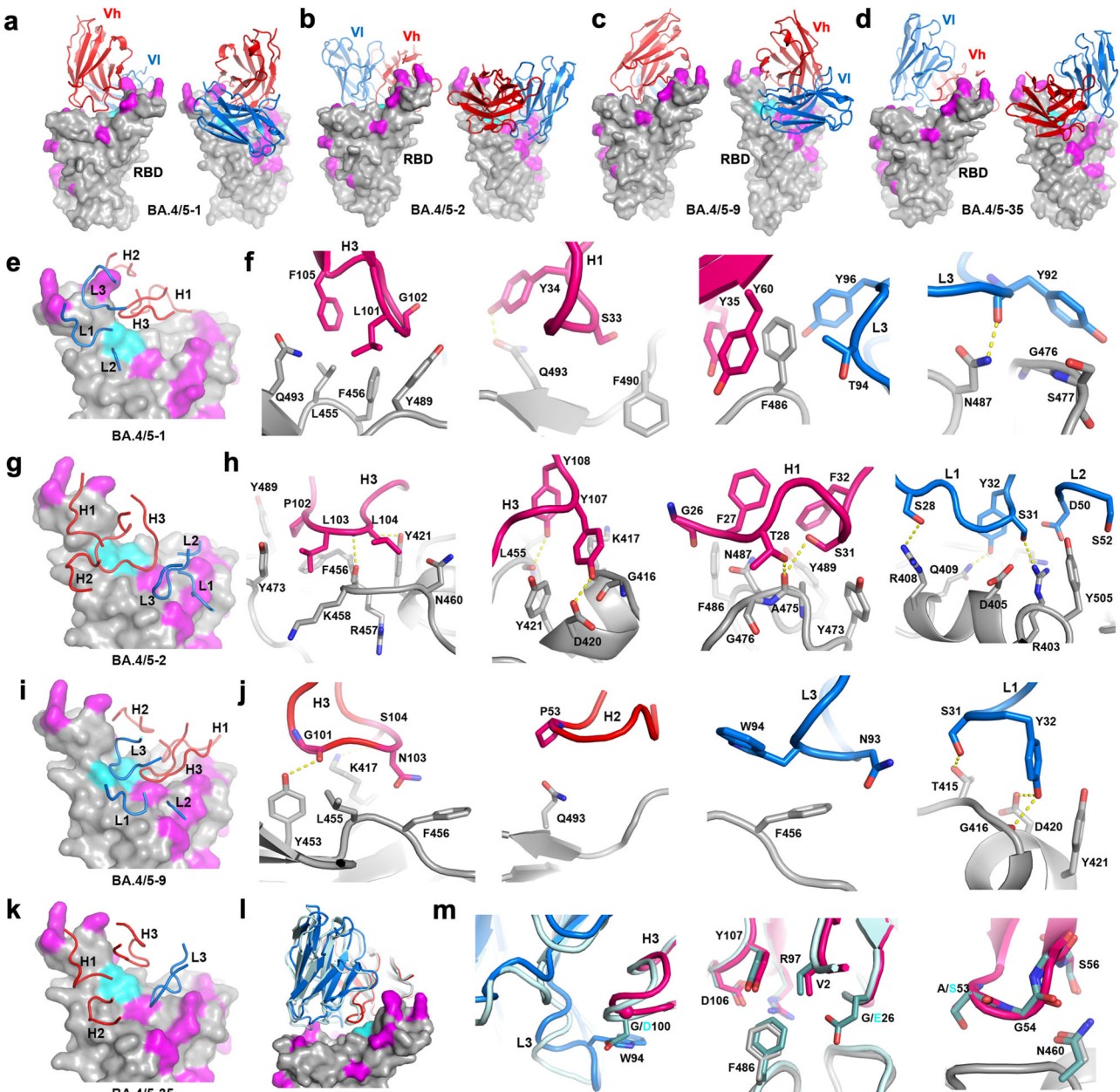

**Fig. 3 | Structures of BA.4/5 Fab complexes with Delta-RBD. a–d** Binding mode of Fabs BA.4/5-1, BA.4/5-2, BA.4/5-9 and BA.4/5−35, respectively, as viewed from the front (left Panel) and back (right panel) of the RBD. RBD is drawn as a grey surface representation with BA.4/5 mutation sites are shown in magenta and residues L455 and F456 in cyan. VhVl domains of the bound Fabs are shown as ribbons and coils with HC in red and LC in blue. **e** and **f** Binding position of the CDRs and detailed interactions for BA.4/5-1/RBD, **g** and **h** for BA.4/5-2/RBD, and **i** and **j** for BA.4/5-9/ RBD. Protein main chains are drawn as ribbons and coils, and side chains as sticks with Fab HC in red and LC in blue, RBD in grey. **k** Binding position of the CDRs for BA.4/5-35/RBD. **l** and **m** Binding mode and detailed interactions of BA.4/5-35 (HC in red and LC in blue) compared with Omi-3[22] (cyan).

mutations and likely evolved from BA.2 in a chronically infected immunosuppressed host[24].

In this paper we have generated a panel of 28 potent human monoclonal antibodies from BA.4 or BA.5 infected volunteers. 26 mAb binds to the RBD and 2 to the NTD. Since the emergence of BA.4/5 SARS-CoV-2, it has continued to evolve antigenically to the point now where variants such as XBB.1.5.70 have developed resistance or complete knockout of activity to all 28 mAb. To achieve this, mutations are targeted at the sites of binding of potent mAbs, most of which bind either on or in close proximity to the ACE2 interacting surface of the RBD and are exemplified by the Fab/RBD structures we present for four representative mAbs.

The continued evolution of SARS CoV-2 in the face of vaccination and frequently multiple rounds of natural infection has essentially allowed the virus to avoid pre-existing immunity (antibody responses found for >95% of adults in the UK (https://www.ons.gov.uk/ peoplepopulationandcommunity/healthandsocialcare/conditionsand diseases/articles/coronaviruscovid19latestinsights/antibodies)), placing the virus under extreme selective pressure. Figure 2b shows that in response, 16% of the residues of the RBD have mutated, corresponding to 26% of the accessible surface of the RBD and over half (56%) of the area of the ACE footprint has been mutated in the drive to escape ACE2-blocking antibodies and optimize ACE2 binding. All monoclonal antibodies selected for prophylactic or therapeutic use bind to the RBD,

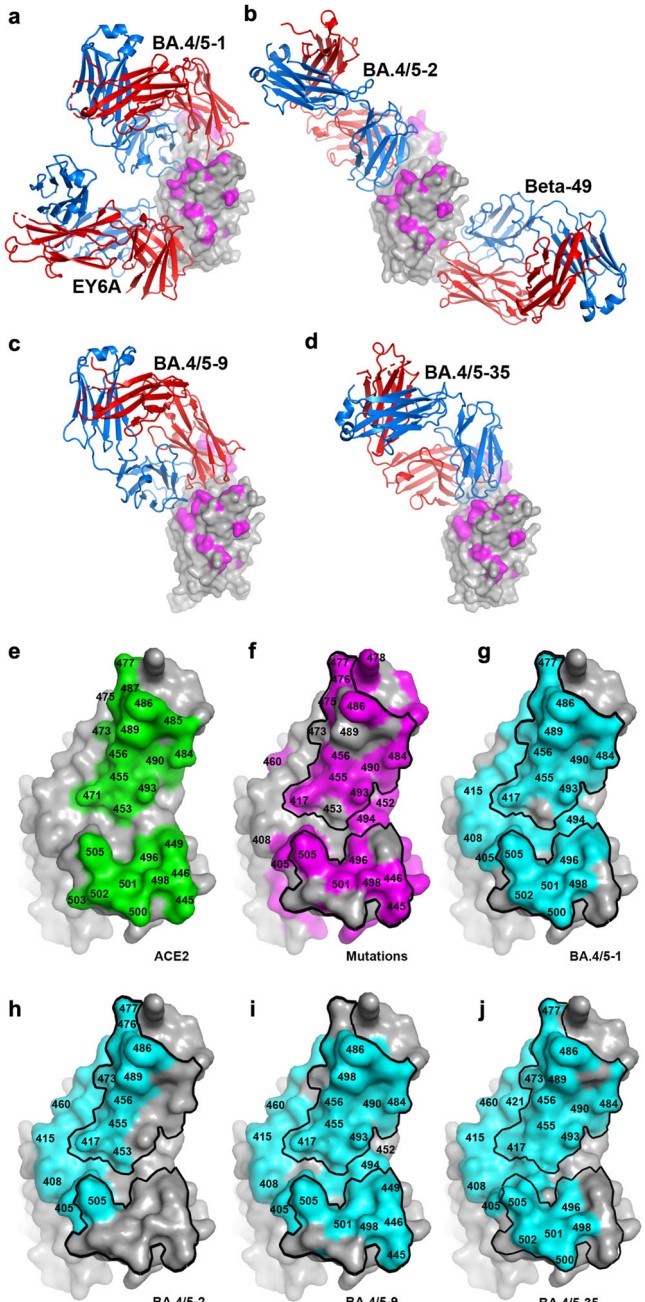

**Fig. 4 | Structures of complexes, ACE2 footprint, mutation sites and Fab footprints on RBD. a–d** Side views (relative to human torso analogy[7]) of RBDs for the crystal structures of Delta-RBD/BA.4/5-1/EY6A[20], Delta-RBD/BA.4/5-2/Beta-49[21], Delta-RBD/BA.4/5-9 and Delta-RBD/BA.4/5-35, respectively. RBD is shown as a grey surface representation with BA.4/5 mutation sites in magenta, Fabs as ribbons with HC in red and LC in blue. **e–j** Surface representation of RBD viewed from the top **e** Residues on the ACE2 footprint shown in green. **f** RBD with all the mutation sites of variants tested against BA.4/5 mAbs in magenta and ACE2 footprint marked by a black outline. **g–j** BA.4/5-1, BA.4/5-2, BA.4/5-9 and BA.4/5-35 footprints are shown in cyan, respectively, and the ACE2 footprint is marked by a black outline.

some were derived from mAbs isolated following SARS-CoV-1 infection (S309 and ADG20)[11,25], whilst others were isolated from cases infected with SARS-CoV-2 during the early pandemic. These first-generation SARS-CoV-2 mAbs bind to areas of the RBD that were hotspots for neutralizing antibody binding and, therefore, also hotspots for mutational escape.

It is, therefore, unsurprising that the evolution of SARS-CoV-2 has led to the steady attrition of the available prophylactic and therapeutic mAbs. Nevertheless, it is remarkable that in 24 months since the emergence of BA.4/5, variants have accrued mutations to knock out the activity of all potent mAbs isolated from cases suffering vaccine breakthrough BA.4/5 infections. The final blow came from the acquisition of mutations at residues 455 and 456, which lie at the back of the left shoulder of the RBD[7], this region is used by the public antibodies belonging to IGHV3-53/66 and a number of other potent ACE2-blocking antibodies, which are consistently knocked-out by this pair of mutations. The cumulative result is that none of the BA.4/5 antibodies described here remain effective. Since this region of the RBD has been resistant to change until recently[22], neutralizing responses have become increasingly focussed on it. This would explain the recurrence of mutations in this region in recent variants. Indeed they are now present in the three most frequent variants circulating within the past 3 months, JN.1 (F455S 13.76%), HV.1 (F456L 12.39%), HK.3 (L455F + F456L 6.49%), reflecting the intense selective pressure on the virus to evade antibodies binding to this region[23]. In the past three months, there have been 196763 sequences submitted to the covSPECTRUM database        (https://cov-spectrum.org/explore/World/AllSamples/ Past3M/variants?&) and F456L accounted for 54.84% (107908) of them, while L455F accounted for 27.49% (54084). The recently described variant BA.2.86 lacks mutations at residues 455 and 456, but it is notable that the L455S mutation has been found in 46610 deposited sequences (the most common subvariant is JN.1) and the F456L mutation in 125 deposited sequences and these may therefore possess a selective advantage over BA.2.86 in the coming months.

In summary, these findings show the remarkable ability of SARS-CoV-2 to accrue mutations to escape pre-existing antibody responses leading in the space of just two years to the emergence of variants such as XBB.1.5.70 containing the FLip mutation able to evade all the potent mAb generated following BA.4/5 infection. It remains to be seen how much of the potential antigenic space has been explored by SARS-CoV-2, but at this point it seems likely that there are many mutations and combinations of mutations yet to be explored. At first glance, it may have been predicted that the ACE2 binding surface may be resistant to mutation due to the need to maintain affinity to ACE2, meaning that antibodies binding to this site would be difficult to escape. To the contrary, the virus has demonstrated that the ACE2 binding surface is far from an Achilles heel, in fact, it shows great plasticity allowing mutational changes to be introduced, which destroy the binding sites for potent antibodies whilst maintaining sufficient ACE2 affinity to allow productive infection.

## Methods

### Ethics statement

Our research complies with all relevant ethical regulations, our research samples were co-enrolled into one or more of the following three studies: the ISARIC/WHO Clinical Characterization Protocol for Severe Emerging Infections [Oxford REC C, reference 13/SC/0149], the "Innate and adaptive immunity against SARS-CoV-2 in healthcare worker family and household members" protocol (approved by the University of Oxford Central University Research Ethics Committee), or the Gastro-intestinal illness in Oxford: COVID sub-study [Sheffield REC, reference: 16/YH/0247].

### Bacterial strains and cell culture

Vero (ATCC CCL-81) and VeroE6/TMPRSS2 cells were cultured in Dulbecco's Modified Eagle medium (DMEM) high glucose (Sigma-Aldrich) supplemented with 10% foetal bovine serum (FBS), 2 mM GlutaMAX (Gibco, 35050061) and 100 U/ml of penicillin–streptomycin at 37 °C. Human mAbs were expressed in HEK293T cells cultured in FreeStyle™ 293 Expression Medium (Cat# 12338018, Gibco™) at 37 °C with 5% $CO_2$.

HEK293T (ATCC CRL-11268) cells were cultured in DMEM high glucose (Sigma-Aldrich) supplemented with 10% FBS, 1% 100X Mem Neaa (Gibco) and 1% 100X L-Glutamine (Gibco) at 37 °C with 5% CO$_2$. To express RBD, RBD variants and ACE2, HEK293T cells were cultured in DMEM high glucose (Sigma) supplemented with 2% FBS, 1% 100X Mem Neaa and 1% 100X L-Glutamine at 37 °C for transfection. BA.5 RBD was expressed in HEK293T (ATCC CRL-11268) cells cultured in FreeStyle™ 293 Expression Medium (Cat# 12338018, Gibco™) at 37 °C with 5% CO$_2$. *E.coli DH5α* bacteria were used for transformation and large-scale preparation of plasmids. A single colony was picked and cultured in LB broth at 37 °C at 200 rpm in a shaker overnight. To produce pseudo-typed lentivirus, HEK293T/17 cells were cultured in Dulbecco's Modified Eagle medium (DMEM) high glucose (Sigma-Aldrich) supplemented with 10% foetal bovine serum (FBS), 2 mM GlutaMAX (Gibco, 35050061) and 100 U/ml of penicillin–streptomycin at 37 °C.

### Sera and PBMC from BA.4/5 infected cases, study subjects

Following informed consent, individuals with Omicron BA.4 or BA.5 were co-enroled into one or more of the following three studies: the ISARIC/WHO Clinical Characterization Protocol for Severe Emerging Infections [Oxford REC C, reference 13/SC/0149], the "Innate and adaptive immunity against SARS-CoV-2 in healthcare worker family and household members" protocol (approved by the University of Oxford Central University Research Ethics Committee), or the Gastro-intestinal illness in Oxford: COVID sub study [Sheffield REC, reference: 16/YH/0247]. Diagnosis was confirmed through reporting of symptoms consistent with COVID-19, hospital presentation, and a test positive for SARS-CoV-2 using reverse transcriptase polymerase chain reaction (RT-PCR) from an upper respiratory tract (nose/throat) swab tested in accredited laboratories and lineage sequence confirmed through national reference laboratories in the United Kingdom. A blood sample was taken following consent at least 14 days after PCR test confirmation. Clinical information, including the severity of the disease (mild, severe or critical infection according to recommendations from the World Health Organization), times between symptom onset and sampling and the age of participants was captured for all individuals at the time of sampling. Sex and gender were not considered in the study design and sex of participants was determined based on self-report. Here we are not evaluating the impact of sex in immune response.

### Isolation of BA.4/5 S-specific or BA.4+all S-specific single B cells by FACS

BA.4/5 S-specific or BA.4+all S-specific single B cell sorting was performed as previously described[7,26]. Briefly, PBMC were stained with LIVE/DEAD Fixable Aqua dye (Invitrogen) followed by recombinant trimeric S-twin-Strep of BA.4/5 or BA.4+all. Cells were then incubated with CD3-FITC (1:10 dilution, BD, Cat. 555332), CD14-FITC (1:10 dilution, BD, Cat 555397), CD16-FITC (1:10 dilution, BD, 555406), CD56-FITC (1:50 dilution, BD, Cat. 562793), IgM-FITC (1:10 dilution, BD, Cat. 555782), IgA-FITC (1:50 dilution, Dako, Cat. F0188), IgD-FITC (1:10 dilution, Dako, Cat. F0189), IgG-BV786 (1:20 dilution, BD, Cat. 564230) and CD19-BUV395 (1:50 dilution, BD, Cat. 563549), along with Strep-MABS-DY549 (1:50 dilution, iba, Cat. 2-1566-050) to stain the twin strep tag of the S protein. IgG+ memory B cells were gated as CD19+, IgG+, CD3−, CD14−, CD56−, CD16−, IgM−, IgA− and IgD−, and S+ were further selected, and single cells were sorted into 96-well PCR plates with 10 μl of catching buffer (Tris, Nuclease free-H$_2$O and RNase inhibitor). Plates were briefly centrifuged at 2000×*g* for 1 min and left on dry ice before being stored at −80 °C.

### Cloning and expression of BA.4/5 S-specific and BA.4+all S-specific human mAbs

BA.4/5 S-specific and BA.4+all S-specific human mAbs were cloned and expressed as described previously[7]. Briefly, genes for Ig IGHV, Ig Vκ and Ig Vλ were recovered from positive wells by RT-PCR. Genes encoding Ig IGHV, Ig Vκ and Ig Vλ were then amplified using Nested-PCR by a cocktail of primers specific to human IgG (see Supplementary Data 1). PCR products of HCs and LCs were ligated into the expression vectors of human IgG1 or immunoglobulin κ-chain or λ-chain by Gibson assembly[27]. For mAb expression, plasmids encoding HCs and LCs were co-transfected by PEI-transfection into a HEK293T cell line, and supernatants containing mAbs were collected, filtered 4-5 days after transfection, and the supernatants further characterized or purified.

### ACE2 binding inhibition assay by ELISA

MAXISORP immunoplates were coated with 5 μg/ml of purified ACE2-His protein overnight at 4 °C and then blocked by 2% BSA in PBS. Meanwhile, mAbs were serially diluted and mixed with 2.5 μg/ml of recombinant BA.4/5 trimeric S-twin-Strep. Antibody-S protein mixtures were incubated at 37 °C for 1 h. After incubation, the mixtures were transferred into the ACE2-coated plates and incubated for 1 h at 37 °C. After washing, StrepMAB-Classic (2-1507-001, iba) was diluted at 0.2 μg/ml by 2% BSA and used as the primary antibody, followed by Goat anti-mouse IgG-AP (A9316, Sigma-Aldrich) at 1:10,000 dilution. The reaction was developed by adding PNPP substrate and stopped with NaOH. The absorbance was measured at 405 nm. The ACE2/S binding inhibition was calculated by comparing it to the antibody-free control well. IC50 was determined using the Probit programme from the SPSS package.

### Focus reduction neutralization assay (FRNT)

The neutralization potential of BA.4/5 infected serum samples was measured using a Focus Reduction Neutralization Test (FRNT), where the reduction in the number of the infected foci is compared to a negative control well without serum. Briefly, serially diluted serum was mixed with SARS-CoV-2 strains and incubated for 1 h at 37 °C. The mixtures were then transferred to 96-well, cell culture-treated, flat-bottom microplates containing confluent Vero cell monolayers in duplicate and incubated for a further 2 h followed by the addition of 1.5% semi-solid carboxymethyl cellulose (CMC) overlay medium to each well to limit virus diffusion. A focus-forming assay was then performed by staining Vero cells with human anti-NP mAb (mAb206) followed by peroxidase-conjugated goat anti-human IgG (A0170; Sigma). Finally, the foci (infected cells) -100 per well in the absence of antibodies were visualized by adding TrueBlue Peroxidase Substrate. Virus-infected cell foci were counted on the classic AID EliSpot reader using AID ELISpot software. The percentage of focus reduction was calculated and IC$_{50}$ was determined using the probit programme from the SPSS package.

### Pseudovirus plasmid construction and lentiviral particle production

Pseudotyped lentivirus expressing SARS-CoV-2 S proteins from ancestral strain (Victoria, S247R), BA.2, BA.4/5, BA.2.75, BA.2.75.2, BA.2.3.20, BA.2.10.4, BJ.1, BN.1, and BA.4.6 were constructed as described previously[17,18,21,22]. Primers are listed in Supplementary Data 1. We applied the same method to construct BQ.1, BQ.1.1, BS.1, and BF.7 by adding more mutations into the BA.4/5 construct. BA.5.9 was created by adding the R346I mutation into BA.4/5 backbone. To generate BQ.1, we added K444T and N460K into BA.4/5 backbone, we then further introduced R346T into BQ.1 to create BQ.1.1 and added A475V into BQ.1.1 to create BQ.1.1 + A475V. To construct BS.1, we added R346T, L452R, N460K and G476S into BA.2 backbone. XBB was constructed by adding the following mutations into BA.2 backbone: V83A, H146Q, Q183E, R346T, L368I, V445P, G446S, N460K, F486S, F490S, and R493Q. To construct XBB.1, G252V was introduced into XBB, and F486P was added into XBB.1 to make XBB.1.5. Similarly, XBB.1.5.10 was constructed by introducing F456L mutation into XBB.1.5, and XBB.1.5.70 by adding L455F into XBB.1.5.10. XBF was constructed by

adding F486P into BN.1, and then reverse mutated 356T in BN.1 to 356K in the original strain. To create BF.7, R346T was introduced in the BA.4/5 backbone. CH.1.1 was created by adding K444T and L452R into BA.2.12.1 and changed T444 in CH.1.1 to M444 to construct CA.3.1. To create DS.1, R403K and F486S were added into BN.1 template. To create BA.2.86 pseudovirus, the S gene was custom synthesized by Integrated DNA Technologies based on the wild-type SARS-CoV-2 BA.2.86 (EPI_ISL_18110065) and cloned into pcDNA3.1 plasmid. The S gene contains following mutations comparing with the original S: ins16MPLF, T19I, R21T, L24del, P25del, P26del, A27S, S50L, H69del, V70del, V127F, G142D, Y144del, F157S, R158G, N211del, L212I, V213G, L216F, H245N, A264D, I332V, G339H, K356T, S371F, S373P, S375F, T376A, R403K, D405N, R408S, K417N, N440K, V445H, G446S, N450D, L452W, N460K, S477N, T478K, N481K, V483del, E484K, F486P, Q498R, N501Y, Y505H, E554K, A570V, D614G, P621S, H655Y, I670V, N679K, P681R, N764K, D796Y, S939F, Q954H, N969K, P1143L. JN.1 was constructed by introducing L455S and a reverse mutation, I670V, into the BA.2.86S gene construct. The resulting pcDNA3.1 plasmid carrying the S gene was used for generating pseudoviral particles together with the lentiviral packaging vector and transfer vector encoding luciferase reporter.

All constructs were Sanger sequence confirmed.

## Pseudoviral neutralization test
The pseudoviral neutralization test has been described previously[21]. Briefly, the neutralizing activity of potent monoclonal antibodies generated from donors who had recovered from BA.4 and BA.5 infections were tested against Victoria, Alpha, Beta, Gamma, Delta, BA.1, BA.1.1, BA.2, BA.2.12.1, BA.2.75, BA.2.75.2, BA.2.3.20, BA.2.10.4, BJ.1, BA.4/5, BA.4.6, BA.5.9, BQ.1, BQ.1.1, BQ.1.1 + A475V, BS.1, BF.7, BN.1, XBB, XBB.1, XBB.1.5, XBB.1.5.10, XBB.1.5.70, XBF, CH.1.1, CA.3.1, DS.1, BA.2.86, and JN.1. Four-fold serial diluted mAbs were incubated with pseudoviral particles at 37 °C with 5% $CO_2$ for 1 h. Stable HEK293T/17 cells expressing human ACE2 were then added to the mixture at $1.5 \times 10^4$ cells/well. 48 h post infection, culture supernatants were removed and 50 μL of 1:2 Bright-Glo TM Luciferase assay system (Promega, USA) in 1× PBS was added to each well. The reaction was incubated at room temperature for 5 min and firefly luciferase activity was measured using CLARIOstar® (BMG Labtech, Ortenberg, Germany). The percentage neutralization was calculated relative to the control. Probit analysis was used to estimate the dilution that inhibited half maximum pseudotyped lentivirus infection (PVNT50).

## Cloning of spike, RBD and NTD
Expression plasmids encoding BA.4/5 spike and RBD were constructed with human codon-optimized sequence from BA.4/5 spike as previously described[17]. Mutations of G339H, R346T, L368I, K444R, V445P, G446S, N450D, L452M, N460K, V483A, A484R, V486S, F490S and S494P were introduced into BA.4/5 spike and RBD expression plasmids to create BA.4+all spike and BA.4+all RBD. The constructs were verified by Sanger sequencing.

## Protein production
Protein expression and purification were largely the same as described previously[7,28]. Twin-strep tagged BA.4/5 and BA.4+all spikes were transiently expressed in HEK293T cells and purified with Strep-Tactin XT resin (IBA Lifesciences). Plasmids encoding BA.4/5 and BA.4+all RBD were transiently expressed in Expi293F™ Cells (ThermoFisher), cultured in FreeStyle™ 293 Expression Medium (ThermoFisher) at 30 °C with 8% $CO_2$ for 4 days. The harvested medium was concentrated using a QuixStand benchtop system. His-tagged RBDs were purified with a 5 mL HisTrap nickel column (GE Healthcare), followed by a Superdex 75 10/300 GL gel filtration column (GE Healthcare).

## IgG mAb and Fab production
AstraZeneca and Regeneron antibodies were provided by AstraZeneca, Vir, Lilly and Adagio antibodies were provided by Adagio, LY-CoV1404 was provided by LifeArc. For the in-house antibodies, heavy and light chains of the indicated antibodies were transiently transfected into 293T cells and antibody purified from supernatant on protein A as previously described[22]. Fabs were digested from purified IgGs with papain using a Pierce Fab Preparation Kit (Thermo Fisher), following the manufacturer's protocol.

## Crystallization, X-ray data collection and structure determination
Purified Delta RBD was deglycosylated with Endoglycosidase H1. Fabs BA.4/5-1 and EY6A, and BA.4/5-2 and Beta-49 were mixed with Delta RBD separately in a 1:1:1 molar ratio, with a final concentration of 7.0 mg ml$^{-1}$. Initial screening of crystals was set up in Crystalquick 96-well X plates (Greiner Bio-One) with a Cartesian Robot using the nanoliter sitting-drop vapour-diffusion method, with 100 nL of protein plus 100 nL of reservoir in each drop, as previously described[29]. Crystals of Delta-RBD/BA.4/5-1/EY6A[20] were formed in Hampton Research PEGRx condition 1-17, containing 0.1 M sodium citrate tribasic dihydrate pH 5.5 and 22% (w/v) PEG 1000. Crystals of Delta-RBD/BA.4/5-2/Beta-49[21] were formed in Hampton Research PEGRx condition 1-46, containing 0.1 M sodium citrate tribasic dihydrate pH 5.0 and 18% (w/v) PEG 20000. Crystals of Delta-RBD/BA.4/5-9 were grown in conditions containing 0.1 M Bis–Tris pH 6.5, 2% v/v polyethylene glycol monomethyl ether 550 and 1.8 M ammonium sulfate. Crystals of Delta-RBD/BA.4/5-35 were grown in 0.1 M sodium citrate tribasic dihydrate pH 5.5, 18% (w/v) PEG 3350.

Crystals were mounted in loops and dipped in a solution containing 25% glycerol and 75% mother liquor for a second before freezing in liquid nitrogen. Diffraction data of Delta RBD/EY6A/BA.4/5-1 and Delta RBD/Beta-49/BA.4/5-2 were collected at beamline i04, and Delta-RBD/BA.4/5-9 and Delta-RBD/BA.4/5-35 at i03 of Diamond Light Source, UK, using the automated queue system that allows unattended automated data collection (https://www.diamond.ac.uk/Instruments/Mx/I03/I03-Manual/Unattended-Data-Collections.html). 3600 diffraction images of 0.1° each were collected at 100 K from a single crystal for each data set. Data integration, scaling and reduction were automatically done with Xia2-dials[30,31]. The structures were determined using molecular replacement with Phaser[32], model rebuilding done with COOT[33] and refinement with Phenix[34]. Data collection and structure refinement statistics are given in Table S2. Structural comparisons used SHP[35] and figures were prepared with PyMOL (The PyMOL Molecular Graphics System, Version 1.2r3pre, Schrödinger, LLC).

## Statistics and reproducibility
Throughout these analyses, the Wilcoxon matched-pairs signed rank test was used as appropriate and two-tailed $P$ values were calculated (see figure legends). No statistical method was used to predetermine the sample size. No data were excluded from the analyses. The experiments were not randomized. The Investigators were not blinded to allocation during experiments and outcome assessment.

## Reporting summary
Further information on research design is available in the Nature Portfolio Reporting Summary linked to this article.

## Data availability
Coordinates have been deposited with the PDB: Delta-RBD/BA.4/5-1/EY6A, 8CBD; Delta-RBD/BA.4/5-2/Beta-49, 8CBE; Delta-RBD/BA.4/5-9, 8QZR; Delta-RBD/BA.4/5-35, 8CMA. Source data are provided with this paper.

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

## Acknowledgements

This work was supported by the Chinese Academy of Medical Sciences (CAMS) Innovation Fund for Medical Science (CIFMS), China (grant number: 2018-I2M-2-002) to D.I.S. and G.R.S. We are also grateful for support from the Red Avenue Foundation and the Oak Foundation. G.R.S. J.M. and D.I.S. are supported by Wellcome through the SEA-COVARIANTS consortium. J.Ren is supported by Wellcome (101122/Z/13/Z), D.I.S. and E.E.F. by the UKRI MRC (MR/N00065X/1) and HMED by the Wellcome Trust Core Award Grant (203141/Z/16/Z). D.I.S. and G.R.S. are Jenner Investigators. This is a contribution from the UK Instruct-ERIC Centre. A.J.M. is an NIHR-supported Academic Clinical Lecturer. The convalescent sampling was supported by the Medical Research Council grant MC_PC_19059, the National Institutes for Health and Oxford Biomedical Research Centre and an Oxfordshire Health Services Research Committee grant to A.J.M. The Wellcome Centre for Human Genetics is supported by the Wellcome Trust (grant 090532/Z/09/Z). The computational aspects of this research were supported by the Wellcome Trust Core Award Grant Number 203141/Z/16/Z and the NIHR Oxford BRC. The Oxford Vaccine work was supported by UK Research and Innovation, Coalition for Epidemic Preparedness Innovations, National Institute for Health Research (NIHR), NIHR Oxford Biomedical Research Centre, Thames Valley and South Midland's NIHR Clinical Research Network. We thank the Oxford Protective T-cell Immunology for COVID-19 (OPTIC) Clinical team for participant sample collection and the Oxford Immunology Network Covid-19 Response T-cell Consortium for laboratory support. This work was supported by the UK Department of Health and Social Care as part of the PITCH (Protective Immunity from T cells to Covid-19 in Health workers) Consortium, the UK Coronavirus Immunology Consortium (UK-CIC), the ISARIC consortium (https://isaric4c.net/about/authors/) and the Huo Family Foundation. We thank the staff of the Wellcome Centre for Human Genetics for Flow cytometry cell sorting. P.K. is an NIHR Senior investigator and is funded by WT222426/

Z/21/Z and NIH (U19 IO82360). S.J.D. is funded by an NIHR Global Research Professorship (NIHR300791).

## Author contributions

C.L., A.D.-G., M.S., R.D., G.R.S. and J.M. generated and characterized mAbs. A.D.-G., C.L. and P.S. generated pseudoviruses and performed neutralization tests. N.G.P, M.A.W. and D.R. H. collected X-ray data and J.R., D.Z., H.M.E.D., E.E.F. and D.I.S. analysed the structural results. D.Z. produced and characterized proteins. N.T. provided materials. A.J.M., P.K., T.G.R. and S.J.D. assisted with patient samples and vaccine trials in the OPTIC Healthcare Worker and the Sheffield STH-Obs studies. S.J.D. and P.K. conceived the study of vaccinated healthcare workers and oversaw the study and sample collection/processing. G.R.S. and D.I.S. conceived the study and wrote the initial manuscript draft, with other authors providing editorial comments. All authors read and approved the manuscript.

## Competing interests

G.R.S. sits on the GSK Vaccines Scientific Advisory Board, consults for AstraZeneca and is a founder member of RQ Biotechnology. D.I.S. consults for AstraZeneca. Oxford University holds intellectual property related to SARS-CoV-2 mAbs discovered in G.R.S.'s laboratory. S.J.D. is a Scientific Advisor to the Scottish Parliament on COVID-19. The remaining authors declare no competing interests.
