## [Peer Review File · Nature Communications]

Emerging variants develop total escape from potent monoclonal antibodies induced by BA.4/5 infectionReviewers' Comments:

Reviewer #1:

Remarks to the Author:

The authors isolate SARS-CoV-2 spike binding monoclonal antibodies (mAbs) from vaccinated patients who suffered vaccine break-through infections with Omicron sub lineages BA.4 or BA.5. The authors characterise a set of 28 potent (determined via focus reduction neutralization titer) receptor-binding domain or N-terminal domain binding mAbs by evaluating their ACE2 blocking ability and by virus neutralization assay (against 26 variants seen throughout the pandemic). A subset of these were also structurally characterised via X-ray crystallography.

The paper itself is well-written. However, I am not entirely clear what is novel about this study compared to a very similar one published in Cell Reports in 2022 (doi: 10.1016/j.celrep.2023.112271) by many of the same authors. Clearly the focus is different BA.2 (previous paper) vs. BA.4/5 (here) but otherwise I struggle having the manuscript several times now to identify the significance of this new data. This is not helped by the discussion and concluding remarks which are highly limited and, in the opinion of this reviewer, do not really discuss the data presented in the paper or seek to make broader comparisons to, for example, other viruses which are perennially under 'intense selective pressure'.

The methods are sound but there are only very limited functional analyses of the RBD, Abs/mAbs and complexes themselves. The authors could, for example, conduct thermal stability assays to explore how stable individual RBD trimers are and perhaps then infer how this (in-)stability may impact different virus functions, i.e. fusion efficiency. The authors could also consider exploring the binding affinity of the RBD to individual Abs/mAbs via biophysical techniques, like surface plasmon resonance. This data would support the structural evidence provided in the paper.

The structures themselves are high-quality. Having said that, it would be useful to hear from the authors their justification in placing chain K and J in the 8CBD structure. At 1 rmsd, the map for basically the whole two chains is missing, even at 0.5 rmsd there is little to go on. It would also be useful to know how the authors have validated the puckering, geometry etc. of the oligosaccharides in the structures outside of the PDB validation pipeline. Did they use Privateer, for example? This information, and actually any information on how the authors validated their structures pre-deposition is missing from the methods. Furthermore, the section of the results where the structures are reported is at times very difficult to follow. In particular, going back and forth between Figure 3, Figure 4, and the supplementary figures. For example, for the BA.4/5-1 inclusive structure, we are directed simultaneously to Figure 3A, 3E, 3F, 4C, and Figure S2A. Just in terms of accessibility, for this reviewer, it would be more useful to have all figures related to a single structure co-located and then perhaps a wrap-up summary figure in the discussion/conclusions. This would also more closely reflect how the structural data is presented in the paper, i.e. structure-by-structure.

Separately, I note that the authors provided a PDB report for a cryo-EM structure that, as far as I can tell, is not included in the paper itself (PDB: 8R1D). Given that there is no cryo-EM mentioned in the manuscript, including in the methods, I am working on the assumption that this is an error and so have not read this PDB report in detail. It's possible that this PDB report was misnamed and uploaded in place of the report for 8QZR which, as far as I can tell, is missing. For this reason, it was not possible to review in full the 8QZR structure at this stage though the structure itself appears sound.

The figures are generally-speaking high quality and nicely presented. However, several could be improved. Figure 1A – is the data shown the ones that the authors specifically chose to take forward? Can the authors provide a supplementary figure showing all data points for context? I find line 88-93 helpful in terms of orienting myself with the workflow but this could also be incorporated into Figure 1 to improve ease-of-understanding. Figure 2A/B is very difficult to interpret by virtue of the amount of data shown in a single table. It is very inaccessible. In-text, the authors refer to Abs being anti-

specific targets but if you look directly at Figure 2A/B, we are just looking at a huge table of numbers. Perhaps the authors could indicate which Abs target which antigen and split the tables up in this manner? It would also be useful to see how the different strains are related. This information is covered somewhat in-text but again when you then look at Figure 2A/B, it is very difficult to carry this information over and orient yourself. To simplify Figure 2B, where there are no mutations to a given amino acid residue (for any lineage), does that residue really need to be included? If not, that would reduce the complexity and size of the table significantly. To help interpret structure-related figures (Figures 3 and S2, mainly), the authors should label directly what the different colours mean for different domains at least once. I realize that this is included in the figure legend but it would really help orient oneself when looking at a panel of structures.

On table S2, the number of unique reflections (in addition to number of observations/reflections) should be included for each structure. I am not clear what the line under number of protein atoms is for there is no label (incl. 14, 49, 38, 14). Given the complexity of the structures themselves, it would be useful for the authors to list the number of residues by chain in each structure and identify any glycans/oligosaccharides there are, per structure. Again, I am unclear what the line under B factors for protein corresponds to for there is no label. The authors should ensure that for each major group (protein, ligand, water, ions), average B factors are reported. The label for the bond lengths and angles suggests this is for ligands, ions, and water only but I am not sure that is correct? Lastly, for completeness, this table should include Ramachandran plot figures (and from what program they are calculated, MolProbity in CCP4i2 or online, PDB validation pipeline, etc), including: favoured, allowed, outliers.

Reviewer #2:

Remarks to the Author:

Whilst the work appears to be of high quality and the study is well presented, I am not sure the findings are novel enough for publication in Nature Communications.

Major comments:

- Why was Delta RBD used for the structures? The authors state that 16% of the residues of the RBD have been mutated so far, doesn't this argue for using RBD from newer variants? Why was Omicron S not used?
- BA.4+all S trimer is a synthetic construct absent in nature. What can this tell us? Is there an example of a virus with all of these mutations present in the public databases? On which donor samples was this used and how many Abs were pulled out?
- A take-home message at the end of the discussion is missing, in my opinion, what novel features were discovered and why are these important?

Minor comments:

- How Victoria different to the Wuhan strain?
- Vaccination status should be detailed for study participants, e.g., how long since vaccination per donor? What was the severity of their breakthrough infections? This is known, according to the methods.
- What were the ethnicities, ages, and sexes of the donors?
- As the authors mention potential lineage expansion from ancestral infections, were these donors previously infected?
- How many Abs were cloned from each donor?
- How many Abs were identified using the different probes?
- Fig 1B - % of cells in each gate should be shown. At the moment it looks like almost everything is alive in the PBMC preps. Is this really the case from frozen cells or even ex vivo preps?

- Fig 1D – Although perhaps not a major point of the study, calculations of SHM are dependent on accurate germline annotation of highly polymorphic IGHV genes. Have germline alleles been inferred for these donors, e.g. using germline gene inference? Similarly, all IGHD genes are annotated, but these are notoriously difficult to genotype. How can the authors be sure of their original gene/allele?
- Similarly, it is well known that residues outside the HCDR3, which vary between germline alleles of the same gene, are important for the recognition of different VoCs, but this is not discussed.
- Not sure I agree with the “herd immunity” statements/definition throughout. You can argue that herd immunity will not be reached for SARS-CoV-2 due to continued virus evolution; so far, there is no level of immunity that protects the herd, unlike in other diseases which mutate more slowly.

Reviewer #3:

Remarks to the Author:

In the manuscript titled “Emerging variants develop total escape from potent monoclonal antibodies induced by BA.4/5 infection,” Liu et al. have successfully identified 28 potent anti-spike monoclonal antibodies from human PBMCs. These antibodies progressively lose their neutralizing capabilities as mutations accumulate in subsequent variants. The structures of the four most broadly neutralizing antibodies were solved, and the effects of RBD mutations on binding were analyzed. Overall, the manuscript is well-crafted, and the inclusion of a wide array of variants, particularly JN.1, is highly informative. Nevertheless, I recommend the following enhancements for consideration:

1. Clarification is needed on the vaccines administered to the 11 triple-vaccinated volunteers. Specifically, whether they received three doses of the monovalent Wuhan-like vaccine or a bivalent vaccine including the Omicron component would be crucial information.
2. While the authors recovered 414 antibodies, only 28 are discussed in detail. Inclusion of screening data for all 414 antibodies leading to the selection of the 28 “potent RBD or NTD binding antibodies” would be advantageous. A summary table or figure displaying ELISA binding data, or heavy and light chain usage of the 414 antibodies, could serve as a valuable resource for other researchers.
3. Consider expanding Figure 1A to include more recent Omicron subvariants beyond BA.4/5.
4. In Figure 2A, the classification/groups of the 28 antibodies (epitopes) should be included. This addition would help readers assess the representativeness of these antibodies. Thousands of SARS-CoV-2 monoclonal antibodies have been reported and for most antibodies the loss of activity against new variants is expected.
5. The findings related to the impact of mutations on residues 455 and 456 in broadly neutralizing antibodies are noteworthy. A more in-depth analysis of these effects, based on the structural data, would be highly beneficial.
6. On line 228, please verify if reference 24 is correctly cited, as it does not appear to contain information on XBB.1.5.
7. On line 404, the term “assay” in “focus reduction neutralization assay” should be revised to “test” for FRNT.

Thank you for asking us to revise the paper and we thank the reviewers for their comments, which we address, point-by-point below.

Point-by-point rebuttal:

Reviewer #1 (Remarks to the Author):

The authors isolate SARS-CoV-2 spike binding monoclonal antibodies (mAbs) from vaccinated patients who suffered vaccine break-through infections with Omicron sub lineages BA.4 or BA.5. The authors characterise a set of 28 potent (determined via focus reduction neutralization titer) receptor-binding domain or N-terminal domain binding mAbs by evaluating their ACE2 blocking ability and by virus neutralization assay (against 26 variants seen throughout the pandemic). A subset of these were also structurally characterised via X-ray crystallography.

The paper itself is well-written. However, I am not entirely clear what is novel about this study compared to a very similar one published in Cell Reports in 2022 (doi: [10.1016/j.celrep.2023.112271](https://doi.org/10.1016/j.celrep.2023.112271)) by many of the same authors. Clearly the focus is different BA.2 (previous paper) vs. BA.4/5 (here) but otherwise I struggle having the manuscript several times now to identify the significance of this new data. This is not helped by the discussion and concluding remarks which are highly limited and, in the opinion of this reviewer, do not really discuss the data presented in the paper or seek to make broader comparisons to, for example, other viruses which are perennially under 'intense selective pressure'.

We have reworked the discussion to more comprehensively discuss our data and make broader comparisons.

The methods are sound but there are only very limited functional analyses of the RBD, Abs/mAbs and complexes themselves. The authors could, for example, conduct thermal stability assays to explore how stable individual RBD trimers are and perhaps then infer how this (in-)stability may impact different virus functions, i.e. fusion efficiency. The authors could also consider exploring the binding affinity of the RBD to individual Abs/mAbs via biophysical techniques, like surface plasmon resonance. This data would support the structural evidence provided in the paper.

The neutralisation data are the key results, and we aim to interpret those in terms of the structure, we don't feel that there is justification for performing a whole extra raft of experiments.

The structures themselves are high-quality. Having said that, it would be useful to hear from

the authors their justification in placing chain K and J in the 8CBD structure. At 1 rnsd, the map for basically the whole two chains is missing, even at 0.5 rnsd there is little to go on.

8CBD is the structure of the RBD with BA.4/5-1 and EY6A Fabs. EY6A is simply used as a crystallisation chaperone. There are three complexes in the crystallographic asymmetric unit. Indeed, the density for EY6A (chain K and J) in one of the complexes is weak, but the position and orientation of the Fab can be reasonably modelled at a lower contour level, indicating the position of EY6A.

It would also be useful to know how the authors have validated the puckering, geometry etc. of the oligosaccharides in the structures outside of the PDB validation pipeline. Did they use Privateer, for example?

As is unfortunately common, especially in a lower resolution map, the density for oligosaccharides is generally weak. We therefore simply modelled oligosaccharides using the standard geometry. The difference maps suggest that the fit is reasonable.

This information, and actually any information on how the authors validated their structures pre-deposition is missing from the methods.

Structures were validated using Molprobity embedded in the Phenix package. A sentence has been added in the Methods to indicate this.

Furthermore, the section of the results where the structures are reported is at times very difficult to follow. In particular, going back and forth between Figure 3, Figure 4, and the supplementary figures. For example, for the BA.4/5-1 inclusive structure, we are directed simultaneously to Figure 3A, 3E, 3F, 4C, and Figure S2A. Just in terms of accessibility, for this reviewer, it would be more useful to have all figures related to a single structure co-located and then perhaps a wrap-up summary figure in the discussion/conclusions. This would also more closely reflect how the structural data is presented in the paper, i.e. structure-by-structure.

We agree with the reviewer the structural figures it would be nice to have a structure by structure presentation. However, given the number of the structures reported in this paper, the presence of irrelevant Fabs as crystallization chaperones and the limit on the number and space for the main figures, it is difficult to re-arrange the figures.

Separately, I note that the authors provided a PDB report for a cryo-EM structure that, as far as I can tell, is not included in the paper itself (PDB: 8R1D). Given that there is no cryo-EM mentioned in the manuscript, including in the methods, I am working on the assumption that this is an error and so have not read this PDB report in detail. It's possible that this PDB report was misnamed and uploaded in place of the report for 8QZR which, as far as I can tell, is missing. For this reason, it was not possible to review in full the 8QZR structure at this stage though the structure itself appears sound.

We thank the reviewer for pointing this out. The validation report (PDB:8R1D) was mistakenly uploaded for the structure of Delta-RBD/BA.4/5-9 (8QZR). We have now uploaded the correct report.

The figures are generally-speaking high quality and nicely presented. However, several could be improved. Figure 1A – is the data shown the ones that the authors specifically chose to take forward? Can the authors provide a supplementary figure showing all data points for context? I find line 88-93 helpful in terms of orienting myself with the workflow but this could also be incorporated into Figure 1 to improve ease-of-understanding.

Figure S1A has been added to show the data points of Figure 1A, and the samples highlighted in grey are the ones we selected to sort BA.4/5 mAbs.

Figure 2A/B is very difficult to interpret by virtue of the amount of data shown in a single table. It is very inaccessible. In-text, the authors refer to Abs being anti-specific targets but if you look directly at Figure 2A/B, we are just looking at a huge table of numbers. Perhaps the authors could indicate which Abs target which antigen and split the tables up in this manner? It would also be useful to see how the different strains are related. This information is covered somewhat in-text but again when you then look at Figure 2A/B, it is very difficult to carry this information over and orient yourself. To simplify Figure 2B, where there are no mutations to a given amino acid residue (for any lineage), does that residue really need to be included? If not, that would reduce the complexity and size of the table significantly.

Thank you, Figure 2 A and B have been modified to make them simpler to interpret. For Figure 2A, we have classified the antibodies based on their neutralisation abilities and example key escape mutations (this does not imply that other mutations do not also contribute but the figures would become too complex if these were included). For figure 2B, we have deleted all the common mutations compared with BA.4/5 S, only showing the different ones.

To help interpret structure-related figures (Figures 3 and S2, mainly), the authors should label directly what the different colours mean for different domains at least once. I realize that this is included in the figure legend but it would really help orient oneself when looking at a panel of structures.

Thanks. We have labelled the domains in the top row of Figure 3 to make it clear.

On table S2, the number of unique reflections (in addition to number of observations/reflections) should be included for each structure. I am not clear what the line under number of protein atoms is for there is no label (incl. 14, 49, 38, 14). Given the complexity of the structures themselves, it would be useful for the authors to list the number of residues by chain in each structure and identify any glycans/oligosaccharides there are, per structure. Again, I am unclear what the line under B factors for protein corresponds to for there is no label. The authors should ensure that for each major group (protein, ligand, water, ions), average B factors are reported. The label for the bond lengths and angles suggests this is for ligands, ions, and water only but I am not sure that is correct?

Lastly, for completeness, this table should include Ramachandran plot figures (and from what program they are calculated, MolProbity in CCP4i2 or online, PDB validation pipeline, etc), including: favoured, allowed, outliers.

This is a standard table for crystal structures downloaded from the 'guide to authors' of Nature. We can add all the parameters required by the reviewer if acceptable to Nature Communications. We are sorry that the columns in Table S2 were not properly adjusted so that the average B-factors for atom groups were not correctly aligned. We have corrected this.

Reviewer #2 (Remarks to the Author):

Whilst the work appears to be of high quality and the study is well presented, I am not sure the findings are novel enough for publication in Nature Communications.

Major comments:

- Why was Delta RBD used for the structures? The authors state that 16% of the residues of the RBD have been mutated so far, doesn't this argue for using RBD from newer variants? Why was Omicron S not used?

Omicron was tried (and other variants), however after extensive trials Delta was found to give crystals. The mode of engagement will be the same, so lessons imputed from the structure will be robust

- BA.4+all S trimer is a synthetic construct absent in nature. What can this tell us? Is there an example of a virus with all of these mutations present in the public databases? On which donor samples was this used and how many Abs were pulled out?

BA.4+all S trimer is indeed a synthetic construct incorporating all the major mutations existing at the time into BA.4 S background. Remarkably the protein expressed and purified without problem. The reason for using it on sorting was to try to find antibodies which were not affected by the mutations on BA.4/5 S as well as mutations from S of other significant variants, and we believe we could find more broadly cross-neutralising antibodies in this way. 4 samples, BA.4/5-4, BA.4/5-5, BA.4/5-6, and BA.4/5-10, were used for sorting by BA.4+all S, and 124 Abs were isolated with 3 cross-neutralising antibodies characterised (BA.4/5-33, BA.4/5-34, and BA.4/5-35).

- A take-home message at the end of the discussion is missing, in my opinion, what novel features were discovered and why are these important?

The discussion has been updated

Minor comments:

- How Victoria different to the Wuhan strain?

In relation to this paper, Victoria has a S247R mutation in the NTD of spike compared with Wuhan.

- Vaccination status should be detailed for study participants, e.g., how long since vaccination per donor? What was the severity of their breakthrough infections? This is known, according to the methods.

We have addressed in Table S3. We don't know the severity of symptoms in these patients.

- What were the ethnicities, ages, and sexes of the donors?

Table S3 added to address this. We don't have the ethnicities of these patients.

- As the authors mention potential lineage expansion from ancestral infections, were these donors previously infected?

Not that we are aware of, although it's difficult to say with confidence as asymptomatic infections may have occurred.

- How many Abs were cloned from each donor?

The number of Abs isolated from each donor are listed below:

BA.4 S sorting	
Sample	Number of mAbs isolated
BA.4/5-2	65
BA.4/5-4	37
BA.4/5-5	60
BA.4/5-6	29
BA.4/5-7	39
BA.4/5-9	29
BA.4/5-10	59
SUM	318

BA.4+all S sorting	
Sample	Number of mAbs isolated
BA.4/5-4	34
BA.4/5-5	47
BA.4/5-6	16
BA.4/5-10	27
SUM	124

- How many Abs were identified using the different probes?

318 Abs were isolated using BA.4 S and 124 Abs were isolated using BA.4+all S as shown in the list above.

- Fig 1B - % of cells in each gate should be shown. At the moment it looks like almost everything is alive in the PBMC preps. Is this really the case from frozen cells or even ex vivo preps?

% of cells in each gate has been added. From the first and third sections of Figure 1B, it can be seen that not all PBMCs are live cells, since a certain number died during freezing or the staining process.

- Fig 1D – Although perhaps not a major point of the study, calculations of SHM are dependent on accurate germline annotation of highly polymorphic IGHV genes. Have germline alleles been inferred for these donors, e.g. using germline gene inference? Similarly, all IGHD genes are annotated, but these are notoriously difficult to genotype. How can the authors be sure of their original gene/allele?

The germline gene usages of each antibody were analysed by IMGT/V-QUEST (https://www.imgt.org/IMGT_vquest/input) using the antibody sequences, which we believe is the best way to infer the closest genotype.

- Similarly, it is well known that residues outside the HCDR3, which vary between germline alleles of the same gene, are important for the recognition of different VoCs, but this is not discussed.

Indeed, and there are frequent somatic mutations outside of the CDRs, we feel that adding in such discussions would not significantly add to the paper.

- Not sure I agree with the “herd immunity” statements/definition throughout. You can argue that herd immunity will not be reached for SARS-CoV-2 due to continued virus evolution; so far, there is no level of immunity that protects the herd, unlike in other diseases which mutate more slowly.

We have modified this usage.

Reviewer #3 (Remarks to the Author):

In the manuscript titled “Emerging variants develop total escape from potent monoclonal antibodies induced by BA.4/5 infection,” Liu et al. have successfully identified 28 potent anti-spike monoclonal antibodies from human PBMCs. These antibodies progressively lose their neutralizing capabilities as mutations accumulate in subsequent variants. The

structures of the four most broadly neutralizing antibodies were solved, and the effects of RBD mutations on binding were analyzed. Overall, the manuscript is well-crafted, and the inclusion of a wide array of variants, particularly JN.1, is highly informative. Nevertheless, I recommend the following enhancements for consideration:

1. Clarification is needed on the vaccines administered to the 11 triple-vaccinated volunteers. Specifically, whether they received three doses of the monovalent Wuhan-like vaccine or a bivalent vaccine including the Omicron component would be crucial information.

Information has been shown in Table S3. All vaccines received were Wuhan-like vaccine.

2. While the authors recovered 414 antibodies, only 28 are discussed in detail. Inclusion of screening data for all 414 antibodies leading to the selection of the 28 "potent RBD or NTD binding antibodies" would be advantageous. A summary table or figure displaying ELISA binding data, or heavy and light chain usage of the 414 antibodies, could serve as a valuable resource for other researchers.

mAbs were screened first by ELISA then followed by neutralisation assays and only those showed $IC_{50} < 0.1\text{mg/mL}$ against BA.4/5 were selected for sequencing and further studies. Our aim here was to identify neutralising antibodies from breakthrough infection samples, determining their epitopes and evaluating their influence on evolution of SARS-CoV-2 variants. Although it may be valuable to an extent to acquire the information of a large amount of non-neutralising antibodies, it doesn't contribute to the knowledge of the impact of neutralising antibodies on the evolution of the variants.

3. Consider expanding Figure 1A to include more recent Omicron subvariants beyond BA.4/5.

The purpose of Figure 1A was to select suitable candidates for antibody sorting using BA.4/5 S, not to assess the impact of the new variants on BA.4/5 breakthrough infection samples, so we didn't include more recent variants.

4. In Figure 2A, the classification/groups of the 28 antibodies (epitopes) should be included. This addition would help readers assess the representativeness of these antibodies. Thousands of SARS-CoV-2 monoclonal antibodies have been reported and for most antibodies the loss of activity against new variants is expected.

We have modified Figure 2A to classify the antibodies based on their neutralisation abilities and major epitopes.

5. The findings related to the impact of mutations on residues 455 and 456 in broadly neutralizing antibodies are noteworthy. A more in-depth analysis of these effects, based on the structural data, would be highly beneficial.

Agreed these are increasingly interesting, and we have clarified this in the discussion.

6. On line 228, please verify if reference 24 is correctly cited, as it does not appear to contain information on XBB.1.5.

This has been corrected – thank you.

7. On line 404, the term “assay” in “focus reduction neutralization assay” should be revised to “test” for FRNT.

Done

Reviewers' Comments:

Reviewer #1:

Remarks to the Author:

I thank the authors for considering the reviewer comments, and think the paper is in a better state than it was at initial submission. However, by-and-large the points I raised during the initial review period remain.

My principal criticism of this work, and this is something highlighted also by Reviewer 2, revolves around the novelty of the work. I make reference in my initial comments to another other study published by many of the same authors in another journal which makes many of the points highlighted in this submission.

In the initial submission it was very difficult to identify the novelty of the work under consideration here. This was principally because the discussion and conclusions were under-developed. In response to reviewer feedback, the authors have worked on the discussion and I think this is better. The take-home messages for me are: (1) SARS-CoV-2 is capable of accruing mutations to escape pre-existing antibody responses, and (2) the ACE2 binding surface is not precluded from mutation, instead it shows great plasticity.

However, I would argue that these points, in particular (1) have already been made to a lesser-or-greater degree in other publications. Within the NPG family, one could point to Cao et al. (Nature, 2022, doi:10.1038/s41586-022-04980-y), which highlighted nicely that SARS-CoV-2 strains evolve mutations to evade pre-existing antibody responses. Alternatively, Qu et al. (Cell, 2024; doi:10.1016/j.cell.2023.12.026), who explored immune evasion, infectivity, and fusogenicity of the SARS-CoV-2 BA.2.86 and XBB-derived FLip variant. Or, indeed, Djokaite-Guraliuc et al. (Cell Reports, 2023, doi: 10.1016/j.celrep.2023.112271) [a paper by many of the same authors here] which follows a very similar experimental workflow as reported here (though, not a criticism in and of itself) and covers quite a bit of the ground explored in this manuscript.

My view is then that while the data is interesting, the results themselves are not novel or at least they are not presented in a way in which I can identify what is new and what is different about this study (compared to what is already known). I certainly still struggle to identify how this research sits in the broader context of this field. There are limited-to-no comparisons made to other, very similar studies, in the discussion.

I would also question evidence underpinning the final statement of the conclusions: "To the contrary, the virus has demonstrated that the ACE2 binding surface is far from an Achilles heel, in fact it shows great plasticity allowing mutational changes to be introduced which destroy the binding sites for potent antibodies whilst maintaining sufficient ACE2 affinity to allow productive infection."

The authors have demonstrated that mutations to the ACE2 binding surface do not preclude binding by mAbs. However, the authors have not, as far as I can tell conducted any infection experiments so it is entirely conjecture to indicate that these mutations "maintain sufficient ACE2 affinity to allow productive infection".

Of course, if the authors decided to conduct infection experiments to demonstrate clearly that these mutations maintain sufficient ACE2 affinity to allow productive infection, then this would indeed add an element of novelty and certainly round-off a nice study.

Though I appreciate changes made to Figure 2A, 2B, Figure 3, and Supplementary Figure 1A, other comments from my initial review of this manuscript around the use of and accessibility of figures remain. I maintain that having to simultaneously read across Figure 3C, Figure 3I, Figure 3J, Figure 4E, and Supplementary Figure 2C (line 190) is quite difficult and not accessible.

In relation to the structural biology work, I appreciate – acutely – the difficulty in fitting highly mobile domains and carbohydrate moieties into Xray data. With respect to the oligosaccharides, which do typically sit in generally-weak density, I would suggest including the difference maps (both with and without the oligos) in the Supplementary Figures. This way readers can assess for themselves the fit/occupancy.

Reviewer #2:

Remarks to the Author:

The manuscript is much improved after addressing the reviewers' comments and providing clear explanations in the rebuttal. The discussion is improved and the additional technical details help the reader.

A small typo to correct in the abstract, line 42 should read "characterized", not "characterize"

Reviewer #3:

Remarks to the Author:

The authors did not fully utilize the opportunity to significantly enhance their manuscript. Here are the specific concerns:

1. The issue regarding the novelty and significance of the study has not been adequately addressed in the revised manuscript. This remains a critical concern as it directly impacts the perceived value of the study within its field.
2. The manuscript did not incorporate several additional experiments and data that were suggested by the reviewers to fortify the work. While these suggestions were not deemed absolutely necessary, their inclusion could have substantially strengthened the manuscript, particularly in light of concerns about the study's novelty.
3. The revisions made to the manuscript are notably minimal, with less than ten places in the introduction and results sections combined, and only two in the methods section. Although the majority of changes are found in the discussion section, they do not sufficiently address the reviewers' queries for comprehensive responses. Moreover, there appears to be a discrepancy between the changes claimed in the rebuttal letter and those actually present in the manuscript.

Therefore, I would recommend that the authors consider further revising their manuscript to comprehensively address these lingering issues.

Rebuttal

Reviewer #1 (Remarks to the Author):

I thank the authors for considering the reviewer comments, and think the paper is in a better state than it was at initial submission

....

My view is then that while the data is interesting, the results themselves are not novel or at least they are not presented in a way in which I can identify what is new and what is different about this study (compared to what is already known). I certainly still struggle to identify how this research sits in the broader context of this field. There are limited-to-no comparisons made to other, very similar studies, in the discussion.

We feel that the modifications addressed this.

I would also question evidence underpinning the final statement of the conclusions: “To the contrary, the virus has demonstrated that the ACE2 binding surface is far from an Achilles heel, in fact it shows great plasticity allowing mutational changes to be introduced which destroy the binding sites for potent antibodies whilst maintaining sufficient ACE2 affinity to allow productive infection.”

The authors have demonstrated that mutations to the ACE2 binding surface do not preclude binding by mAbs. However, the authors have not, as far as I can tell conducted any infection experiments so it is entirely conjecture to indicate that these mutations “maintain sufficient ACE2 affinity to allow productive infection”.

Of course, if the authors decided to conduct infection experiments to demonstrate clearly that these mutations maintain sufficient ACE2 affinity to allow productive infection, then this would indeed add an element of novelty and certainly round-off a nice study.

The variants studied are naturally occurring and thus infectious.

Though I appreciate changes made to Figure 2A, 2B, Figure 3, and Supplementary Figure 1A, other comments from my initial review of this manuscript around the use of and accessibility of figures remain. I maintain that having to simultaneously read across Figure 3C, Figure 3I, Figure 3J, Figure 4E, and Supplementary Figure 2C (line 190) is quite difficult and not accessible.

As previously stated - given the number of the structures reported in this paper, the presence of irrelevant Fabs as crystallization chaperones and the limit on the number and space for the main figures, it is difficult to re-arrange the figures, however we have now moved a supplementary figure to be part of Figure 4, and making the call outs easier.

In relation to the structural biology work, I appreciate – acutely – the difficulty in fitting highly mobile domains and carbohydrate moieties into Xray data. With respect to the oligosaccharides, which do typically sit in generally-weak density, I would suggest including the difference maps (both with and without the oligos) in the Supplementary Figures. This way readers can assess for themselves the fit/occupancy.

Portions of the maps, including examples with fitted oligosaccharide are now shown in Figure S3.

Reviewer #2 (Remarks to the Author):

The manuscript is much improved after addressing the reviewers' comments and providing clear explanations in the rebuttal. The discussion is improved and the additional technical details help the reader.

A small typo to correct in the abstract, line 42 should read "characterized", not "characterize"

Corrected

Reviewer #3 (Remarks to the Author):

The authors did not fully utilize the opportunity to significantly enhance their manuscript. Here are the specific concerns:

1. The issue regarding the novelty and significance of the study has not been adequately addressed in the revised manuscript. This remains a critical concern as it directly impacts the perceived value of the study within its field.

We believe that the current version adequately addresses these issues

2. The manuscript did not incorporate several additional experiments and data that were suggested by the reviewers to fortify the work. While these suggestions were not deemed absolutely necessary, their inclusion could have substantially strengthened the manuscript, particularly in light of concerns about the study's novelty.

These were beyond the scope of the study.

3. The revisions made to the manuscript are notably minimal, with less than ten places in the introduction and results sections combined, and only two in the methods section. Although the majority of changes are found in the discussion section, they do not sufficiently address the reviewers' queries for comprehensive responses. Moreover, there appears to be a discrepancy between the changes claimed in the rebuttal letter and those actually present in the manuscript.

Therefore, I would recommend that the authors consider further revising their manuscript to comprehensively address these lingering issues.

In word the number of revisions now clocks at 376 for the main text (excluding methods, albeit 71 are formatting changes). This reflects the effort we have made to address this.